# Dynamics of soliton self-injection locking in optical microresonators

Andrey S. Voloshin [1,2,6], Nikita M. Kondratiev [1,6], Grigory V. Lihachev [2,6], Junqiu Liu [2], Valery E. Lobanov [1,3], Nikita Yu. Dmitriev [1,4], Wenle Weng [2], Tobias J. Kippenberg [2✉] & Igor A. Bilenko [1,5✉]

Soliton microcombs constitute chip-scale optical frequency combs, and have the potential to impact a myriad of applications from frequency synthesis and telecommunications to astronomy. The demonstration of soliton formation via self-injection locking of the pump laser to the microresonator has significantly relaxed the requirement on the external driving lasers. Yet to date, the nonlinear dynamics of this process has not been fully understood. Here, we develop an original theoretical model of the laser self-injection locking to a nonlinear microresonator, i.e., nonlinear self-injection locking, and construct state-of-the-art hybrid integrated soliton microcombs with electronically detectable repetition rate of 30 GHz and 35 GHz, consisting of a DFB laser butt-coupled to a silicon nitride microresonator chip. We reveal that the microresonator's Kerr nonlinearity significantly modifies the laser diode behavior and the locking dynamics, forcing laser emission frequency to be red-detuned. A novel technique to study the soliton formation dynamics as well as the repetition rate evolution in real-time uncover non-trivial features of the soliton self-injection locking, including soliton generation at both directions of the diode current sweep. Our findings provide the guidelines to build electrically driven integrated microcomb devices that employ full control of the rich dynamics of laser self-injection locking, key for future deployment of microcombs for system applications.

¹ Russian Quantum Center, Moscow 143026, Russia. ² Institute of Physics, Swiss Federal Institute of Technology Lausanne (EPFL), CH-1015 Lausanne, Switzerland. ³ National University of Science and Technology (MISiS), 119049 Moscow, Russia. ⁴ Moscow Institute of Physics and Technology (MIPT), Dolgoprudny, Moscow Region 141701, Russia. ⁵ Faculty of Physics, M.V. Lomonosov Moscow State University, 119991 Moscow, Russia. ⁶These authors contributed equally: Andrey S. Voloshin, Nikita M. Kondratiev, Grigory V. Lihachev. ✉email: tobias.kippenberg@epfl.ch; i.bilenko@rqc.ru

Recent advances in bridging integrated photonics and optical microresonators[1–4] have highlighted the technological potential of soliton-based microresonator frequency combs ("soliton microcombs")[5–9] in a wide domain of applications, such as coherent communication[10,11], ultrafast optical ranging[12,13], dual-comb spectroscopy[14], astrophysical spectrometer calibration[15,16], low-noise microwave synthesis[17], and to build integrated frequency synthesizers[18] and atomic clocks[19]. Likewise, soliton microcombs also are a testbed for studying the rich nonlinear dynamics, arising from a non-equilibrium driven dissipative nonlinear system, governed by the Lugiato–Lefever equation or extensions thereof, that leads to the formation of "localized dissipative structure"[8,20–25]. To generate soliton microcombs, commonly, the cavity is pumped with a frequency agile, high-power narrow-linewidth, continuous-wave laser with an optical isolator to avoid back reflections. The fast tuning of the laser frequency[26] is applied to access the soliton states, which are affected by the thermal resonator heating. Previously, laser self-injection locking (SIL) to high-Q crystalline microresonators has been used to demonstrate narrow-linewidth lasers[17,27], ultra-low-noise photonic microwave oscillators[28], and soliton microcomb generation[29], i.e., soliton SIL. Microresonators provide a high level of integration with the semiconductor devices, integrated InP-Si₃N₄ hybrid lasers have rapidly become the point of interest for narrow-linewidth on-chip lasers[30–32]. Moreover, 100 mW multi-frequency Fabry–Perot lasers have recently been employed to demonstrate an electrically driven microcomb[33]. Another approach was based on a Si₃N₄ microresonator butt-coupled to a semiconductor optical amplifier (SOA) with on-chip Vernier filters and heaters for soliton initiation and control[34]. The integrated soliton microcomb based on the direct pumping of a Si₃N₄ microresonator by a III–V distributed feedback (DFB) laser has been reported[35,36]. Recent demonstrations of integrated packaging of DFB lasers and ultra-high-Q Si₃N₄ microresonators with low repetition rates[37] made turn-key operation of such devices possible. Low power consumption of integrated microcombs[33] will allow increasing the efficiency of data-centers, which use an estimated 200 TWh each year and are responsible for 0.3% of overall carbon emissions[38].

However, despite the inspiring and promising experimental results, the principles and dynamics of the soliton SIL have not been sufficiently studied yet. Only recently some aspects of the soliton generation effect were investigated[37], where static operation was considered, but comprehensive theoretical and experimental investigation is still necessary. The common SIL models consider either laser equations with frequency-independent feedback[39–41] or linear-resonant feedback[42–44].

Here, we first develop a theoretical model, taking into account nonlinear interactions of the counter-propagating waves in the microresonator, to describe nonlinear SIL, i.e., SIL to a nonlinear microresonator. Using this model, we show that the principles of the soliton generation in the self-injection locked devices differ considerably from the conventional soliton generation techniques. We find that the emission frequency of the laser locked to the nonlinear microresonator is strongly red-detuned and located inside the soliton existence range. For experimental verification, we develop a technique to experimentally characterize the SIL dynamics and study it in a hybrid-integrated soliton microcomb device with 30 GHz repetition rate, amenable to the direct electronic detection, using an InGaAsP DFB laser self-injection locked to a high-quality-factor ($Q_0 > 10^7$) integrated Si₃N₄ microresonator. We demonstrate the presence of the non-trivial dynamics upon diode current sweep, predicted by the theoretical model, and perform the beatnote spectroscopy, i.e., study of the soliton repetition rate evolution under SIL.

## Results

**Principle of laser SIL.** First, we introduce the general principles of the laser SIL to the microresonator (Fig. 1a, b) and clarify definitions of basic terms (Fig. 1c). The generation frequency of the free-running DFB diode is determined by its laser cavity (LC) resonant frequency $\omega_{LC}$ and can be tuned by varying the diode injection current $I_{inj}$ exhibiting practically linear dependence. When $\omega_{LC}$ is tuned into a high-Q resonance of the Si₃N₄ microresonator with frequency $\omega_0$, laser SIL can happen. In that case, $\omega_{eff}$ is the actual or effective laser emission frequency, that differs from the $\omega_{LC}$ as the optical feedback from the microresonator affects the laser dynamics.

It is convenient to introduce the normalized laser cavity to microresonator detuning $\xi = 2(\omega_0 - \omega_{LC})/\kappa$ and the actual effective detuning $\zeta = 2(\omega_0 - \omega_{eff})/\kappa$, where $\omega_0$ is the frequency of the microresonator resonance and $\kappa$ is its loaded linewidth. Note that $\xi$ practically linearly depends on the injection current $I_{inj}$. The $\zeta$ is the actual detuning parameter that determines the dynamics of the nonlinear processes in microresonator[45]. Here we define the "tuning curve" as the dependence of $\omega_{eff}$ on the injection current $I_{inj}$, or equivalently, the dependence of $\zeta$ on $\xi$ (Fig. 1c).

When the laser frequency $\omega_{LC}$ is far detuned from the resonance $\omega_0$ ($|\xi| \gg 0$), the tuning curve first follows the line $\zeta = \xi$ (Fig. 1c) when the injection current $I_{inj}$ changes. When $\omega_{LC}$ is tuned into $\omega_0$, i.e., $\xi \to 0$, the laser frequency becomes locked to the resonance due to the Rayleigh-backscattering-induced SIL, so that $\zeta \approx 0$ despite the variations of $\xi$ within the locking range. The stabilization coefficient (inverse slope at $\xi = 0$) and the locking range $\Delta\omega_{lock}$ are determined by the amplitude and phase of the backscattered light[44]. When $\xi$ increases further and finally moves out of the locking range, the laser becomes free-running again, such that $\zeta = \xi$. Note also, that the locked state region for continuous one-way scanning may not fully coincide with the locking range (see Fig. 1c) and even be different for different scan directions.

Such tuning curves (Fig. 1c and gray curve in Fig. 1d) are well-studied for the case of the linear microresonator (or for small pump powers)[44,46]. However, analyzing experimental results on the soliton formation in the linear SIL regime[29,33], we found that such linear model can't predict soliton generation.

**SIL to a nonlinear microresonator.** Previous works[23,45] have shown that soliton generation occurs in a certain range of the normalized pump detunings $\zeta$, with the lower boundary being above the bistability criterion and the upper boundary being the soliton existence criterion:

$$\zeta \in \left( \left(\frac{f}{2}\right)^{2/3} + \sqrt{4\left(\frac{f}{2}\right)^{4/3} - 1}; \frac{\pi^2 f^2}{8} \right], \tag{1}$$

where $f = \sqrt{8\omega_0 c n_2 \eta P_{in}/(\kappa^2 n^2 V_{eff})}$ is the normalized pump amplitude, $\omega_0$ is the resonance frequency, $c$ is the speed of light, $n_2$ is the microresonator nonlinear index, $P_{in}$ is the input pump power, $n$ is the refractive index of the microresonator mode, $V_{eff}$ is the effective mode volume, $\kappa$ is the loaded resonance linewidth, and $\eta$ is the coupling efficiency ($\eta = 1/2$ for critical coupling). Note, however, that the lower boundary (bistability) includes also a region of breather existence for high pump power[47], so actual soliton region starts at a bit higher detunings. The linear SIL model[44] predicts that, in the SIL regime, the attainable detuning range of $\zeta \in [-0.7; 0.7]$ in the locked state does not overlap with the soliton existence range despite the sufficient pump power.

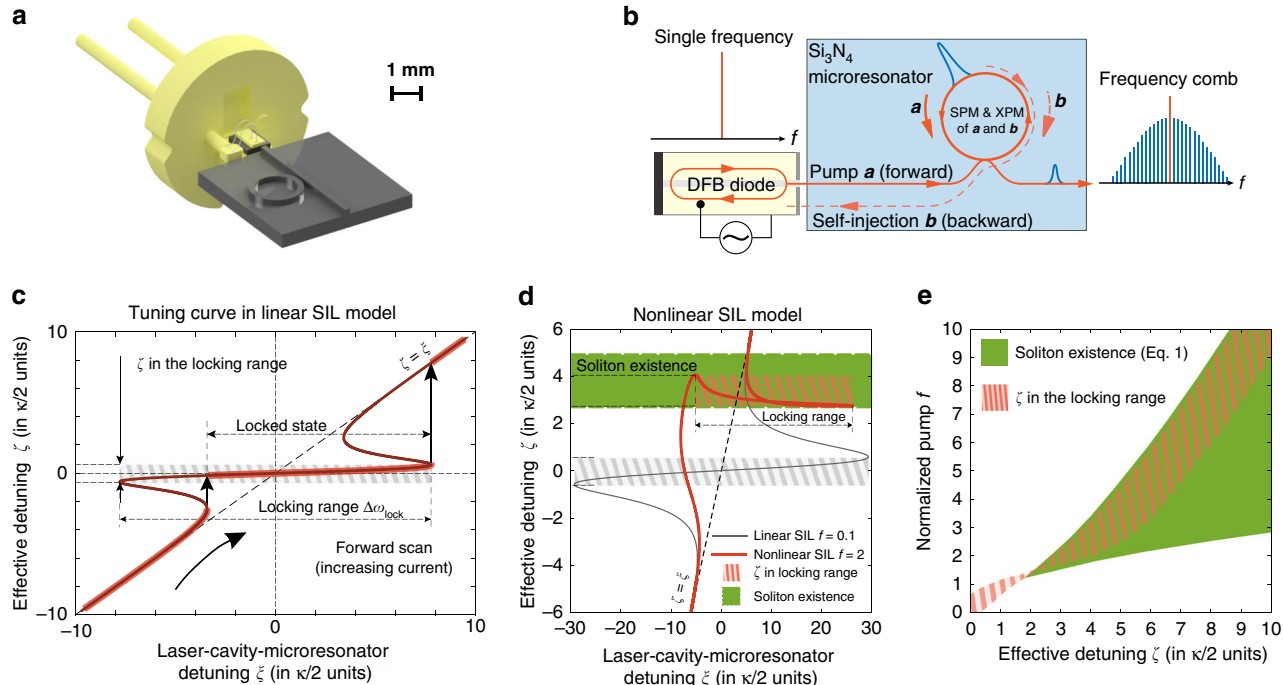

**Fig. 1 Scheme of a compact soliton microcomb using laser self-injection locking. a** Illustration of the soliton microcomb device via direct butt coupling of a laser diode to the Si$_3$N$_4$ chip. **b** Principle of laser self-injection locking. The DFB laser diode is self-injection locked to a high-$Q$ resonance via Rayleigh backscattering and simultaneously pumps the nonlinear microresonator to generate a soliton microcomb. In this work, we introduce and study the influence of the microresonator nonlinearity (self- and cross- phase modulation) on the SIL. Nonlinear SIL model explains the dynamics of the soliton formation in this case. **c** Schematic of the self-injection locking dynamics without taking into account the microresonator nonlinearity, i.e., linear SIL model. The injection current defines the laser cavity frequency $\omega_{LC}$ and the laser cavity-microresonator detuning $\xi = 2(\omega_0 - \omega_{LC})/\kappa \sim l_{inj} - l_0$, while the whole system oscillates at the actual laser emission frequency $\omega_{eff}$, detuned from the cold microresonator at the $\zeta = 2(\omega_0 - \omega_{eff})/\kappa$. We call the dependence of the laser emission frequency on the injection current, or $\zeta$ dependence on $\xi$, a tuning curve. The normalized effective detuning $\zeta$ deviates from $\xi = \zeta$ (free-running case) when self-injection locking occurs. The slope of the tuning curve $d\zeta/d\xi \ll 1$ is observed within the locking range, providing narrowing of the laser diode linewidth. Note, that $\zeta \in [-0.7; 0.7]$ in the locked state for the linear SIL model and is not enough for soliton formation for any pump power. **d** Nonlinear SIL model coincides with the linear one for low pump powers $f < 1$, but the tuning curve changes significantly at higher pump power $f > 1$ and shifts up. **e** Our model predicts that attainable $\zeta$ values in the SIL regime are red-detuned and located inside the soliton existence range (Eq. (1)).

We attribute this contradiction to the absence of the microresonator Kerr nonlinearity in the linear SIL model. The modified nonlinear SIL model including the Kerr nonlinearity[48] is presented as follows.

Consider the microresonator coupled mode equations[45] with backscattering[49] for forward and backward (or clockwise and counter-clockwise propagating) mode amplitudes $a_\mu$ and $b_\mu$, which is analogous to the linear SIL model[44] with additional nonlinear terms:

$$
\begin{aligned}
\dot{a}_\mu = &-(1 + i\zeta_\mu)a_\mu + i\Gamma b_\mu + i\sum_{\mu'=\nu+\eta-\mu} a_\nu a_\eta a_{\mu'}^* + \\
&+ 2i\alpha_x a_\mu \sum_\eta |b_\eta|^2 + f\delta_{\mu0}, \\
\dot{b}_\mu = &-(1 + i\zeta_\mu)b_\mu + i\Gamma a_\mu + i\sum_{\mu'=\nu+\eta-\mu} b_\nu b_\eta b_{\mu'}^* + \\
&+ 2i\alpha_x b_\mu \sum_\eta |a_\eta|^2,
\end{aligned}
\tag{2}
$$

where $\Gamma$ is the normalized coupling rate between forward and backward modes (mode splitting in units of mode linewidth), $\alpha_x$ is a coefficient derived from mode overlap integrals and $\zeta_\mu = 2(\omega_\mu - \mu D_1 - \omega_{eff})/\kappa$ is the normalized detuning between the laser emission frequency $\omega_{eff}$ and the $\mu$-th cold microresonator resonance $\omega_\mu$ on the FSR-grid, with $\mu = 0$ being the pumped mode and $D_1/2\pi$ is the microresonator free spectral range (FSR). For numerical estimations we use $\alpha_x = 1$ as for the modes with

the same polarization. Equation (2) provides a nonlinear resonance curve and the soliton solution[45,49]. For analysis of the SIL effect we combine Eq. (2) with the standard laser rate equations similar to the Lang–Kobayashi equations[39], but with resonant feedback[44]. The pumped mode corresponding to $\mu = 0$ is of main interest. We search for the stationary solution:

$$
\begin{aligned}
-(1 + i\zeta)a + i\Gamma b + ia(|a|^2 + 2\alpha_x|b|^2) + f = 0, \\
-(1 + i\zeta)b + i\Gamma a + ib(|b|^2 + 2\alpha_x|a|^2) = 0,
\end{aligned}
\tag{3}
$$

where we define $a = a_0$, $b = b_0$ and $\zeta = \zeta_0$ for simplicity. These equations define the complex reflection coefficient of the WGM microresonator which is used for SIL theory. To solve Eq. (3) and make resemblance to the linear case, we introduce the nonlinear detuning shift $\delta\zeta_{nl}$

$$
\delta\zeta_{nl} = \frac{2\alpha_x + 1}{2}(|a|^2 + |b|^2)
\tag{4}
$$

and nonlinear coupling shift $\delta\Gamma_{nl}$

$$
\delta\Gamma_{nl} = \frac{2\alpha_x - 1}{2}(|a|^2 - |b|^2)
\tag{5}
$$

We further transform $\bar{\zeta} = \zeta - \delta\zeta_{nl}$, $\bar{\Gamma}^2 = \delta\Gamma_{nl}^2 + \Gamma^2$, in order to achieve Eq. (3) in the same form as in the linear SIL model[44,50]. After redefinition $\bar{\xi} = \xi - \delta\zeta_{nl}$, the nonlinear tuning curve in the

new coordinates $\bar{\xi}$-$\bar{\zeta}$ becomes:

$$\bar{\xi} = \bar{\zeta} + \frac{K_0}{2} \frac{2\bar{\zeta}\cos\bar{\psi} - (1 + \bar{\Gamma}^2 - \bar{\zeta}^2)\sin\bar{\psi}}{\left(1 + \bar{\Gamma}^2 - \bar{\zeta}^2\right)^2 + 4\bar{\zeta}^2}, \qquad (6)$$

where $K_0 = 8\eta\Gamma\kappa_{do}\sqrt{1 + \alpha_g^2}/(\kappa R_o)$ is the SIL coefficient and $\bar{\psi} = \psi_0 - \kappa\tau_s\zeta/2$ is the self-injection phase, $\kappa_{do}$ is the laser diode output mirror coupling rate, $\alpha_g$ is the Henry factor[51] and $R_o$ is the amplitude reflection coefficient of the laser output facet. We also note that the laser cavity resonant frequency $\omega_{LC}$, as well as $\xi$, are also assumed to include the Henry factor in its definition. The $\kappa\tau_s/2$ is usually considered to be small, i.e., $\kappa\tau_s/2 \ll 1$, so the locking phase $\bar{\psi} \approx \psi_0 = \omega_0\tau_s - \arctan\alpha_g - 3\pi/2$ depends on both the resonance frequency $\omega_0$ and the round-trip time $\tau_s$ from the laser output facet to the microresonator and back. The SIL coefficient $K_0$ is analogous to the feedback parameter $C$ used in the theory of the simple mirror feedback[40,41], where the SIL is achieved with the frequency-independent reflector forming an additional Fabry-Perot cavity. However, in the resonant feedback setup the SIL coefficient does not depend on the laser-to-reflector distance, depending on the parameters of the reflector instead. Though the system has qualitatively similar regimes as the simple one[52], their ranges and thresholds are different[42,44,46]. The value of $K_0 > 4$ is required for the pronounced locking with sharp transition, naturally becoming a locking criterion. For high-Q microresonators this value can be no less than several hundred. We also note that in the linear regime (or in nonlinearly shifted coordinates $\bar{\xi}$, $\bar{\zeta}$) the stabilization coefficient of the setup is close to $K_0$, full locking range is close to $0.65K_0 \times \kappa/2$. The nonlinear detuning and coupling can be expressed as

$$\delta\zeta_{nl} = \frac{2\alpha_x + 1}{2}f^2\frac{1 + (\bar{\zeta} - \delta\Gamma_{nl})^2 + \Gamma^2}{\left(1 + \bar{\Gamma}^2 - \bar{\zeta}^2\right)^2 + 4\bar{\zeta}^2}, \qquad (7)$$

$$\delta\Gamma_{nl} = \frac{2\alpha_x - 1}{2}f^2\frac{1 + (\bar{\zeta} - \delta\Gamma_{nl})^2 - \Gamma^2}{\left(1 + \bar{\Gamma}^2 - \bar{\zeta}^2\right)^2 + 4\bar{\zeta}^2}. \qquad (8)$$

Equations (6)–(8) can be solved numerically and plotted in $\zeta = \bar{\zeta} + \delta\zeta_{nl}$, $\xi = \bar{\xi} + \delta\zeta_{nl}$ coordinates[48]. We observe that calculated tuning curve in the nonlinear case, when Kerr nonlinearity is present, differs drastically from the tuning curve predicted by the linear model (cf. red and gray lines in Fig. 1d). Also, we can see from Eq. (7) that the nonlinear detuning shift is positive, and allows for larger detuning $\zeta$ (proportional to the pump power). The proposed nonlinear SIL model is valid for both anomalous and normal group velocity dispersions.

We show the distinctions between conventional generation of the dissipative Kerr solitons and self-injection locked soliton excitation in Fig. 2. Figure 2a, b shows the conventional case where the laser pumps the microresonator with an optical isolator between the laser and the microresonator, preventing SIL. In Fig. 2a, the solid black line corresponds to the linear tuning curve ($\zeta = \xi$) of a free-running laser. In Fig. 2b, the thick solid black curve corresponds to the solution of Eq. (2) in the $\xi$ frame, which provides soliton solutions[49]. Horizontal dashed green lines are the boundaries of the soliton existence range in the $\zeta$ frame, i.e., Eq. (1), highlighted also with the green area.

Next, we consider tuning curves corresponding to the nonlinear SIL, described by Eq. (6). They are plotted in Fig. 2c, e in the $\zeta$–$\xi$ frame with dashed red lines. Due to the multistability of the tuning curves, forward and backward laser scans within the same diode current range result in different ranges of the effective detuning $\zeta$ and different tuning curves (thick solid lines). Figure 2c, e shows the attainable values of detunings $\zeta$, and Fig. 2d, f shows the

intracavity power $|a(\xi)|^2$ for the forward and backward scans (thick dashed lines). We studied a set of the real-world parameters and found that the following key conclusions can be done: first, the effective detuning $\zeta$ predominantly locks to the red-detuned region, where the soliton microcomb formation is possible (Fig. 1e shows the union of the SIL regions for different locking phases together with the soliton existence region). Second, in the SIL regime soliton generation may be observed for both directions of the current sweep (see Fig. 2c, e) that is impossible for the free-running laser. Also, it is possible to obtain larger values of the detuning $\zeta$ in the SIL regime using backward tuning (cf. Fig. 2c, e). At the same time, the locked state width can be shorter for the backward tuning than that for the forward tuning. Third, while decreasing the diode current (i.e., backward tuning, the free-running laser frequency rising), the detuning $\zeta$ can grow (Fig. 2e) that is counter-intuitive. Moreover, such non-monotonic behavior of $\zeta$ may take place in the soliton existence domain and may affect soliton dynamics. As it was shown in[53] decrease of the detuning value in the soliton regime may lead to the switching between different soliton states.

More study of the nonlinear tuning curves dependence on the locking phase $\psi_0$, pump power, and the mode-coupling parameter $\Gamma$ is presented in Supplementary Note 2.

**Dynamics of soliton SIL.** For experimental verification, the integrated soliton microcomb device consisting of a semi-conductor laser diode and a high-Q $Si_3N_4$ microresonator chip is developed. In our experiment, we use a commercial DFB laser diode of 120 kHz linewidth and 120 mW output power, which is directly butt coupled to a $Si_3N_4$ chip (Fig. 3a) without using an optical isolator (see "Butt coupling" in Methods).

The $Si_3N_4$ chip is fabricated using the photonic Damascene reflow process[54,55] and features intrinsic quality factor $Q_0$ exceeds $10 \times 10^6$ (see "Silicon nitride chip information" in Methods). We study two different photonic chips, containing microresonators with FSR of 30.6 GHz and 35.4 GHz. Tuning of the injection current makes the laser diode self-injection locked to the microresonator. The Lorentzian linewidth of the SIL laser is 1.1 kHz. Laser phase noise in the free-running and SIL regimes presented in Fig. 3c demonstrates the laser linewidth reduction by more than 100 times (see Supplementary Note 1 for details). At some particular currents soliton states are generated (see "Comb generation in the SIL regime" in Methods). Figure 3b shows the single soliton spectrum with 30.6 GHz repetition rate. Soliton repetition beatnote signal[56–60] and corresponding phase noise are shown in Fig. 3d (see Supplementary Note 5 for details). The phase noise for the 35.4 GHz soliton is shown in Fig. 3d as well. This soliton state provides ultra-low noise beatnote signal of −96 dBc at 10 kHz frequency offset, that is comparable to some breadboard implementation without any complicated feedback schemes.

Having access to the stable operation of these soliton states, we prove experimentally (in addition to their ultra-low noise RF beatnote signal (Fig. 3d, inset), optical spectrum (Fig. 3b) and zero background noise) that such optical spectra correspond to the ultra-short pulses representing bright temporal solitons. We perform a frequency-resolved optical gating (FROG) experiment[45]. This corresponds to a second-harmonic generation autocorrelation experiment in which the frequency-doubled light is resolved in spectral domain (see Supplementary Note 6 for details). Reconstructed optical field confirms that we observe temporal solitons with the width less than 1 picosecond. Therefore, SIL is a reliable[33,37] platform to substitute bulky narrow-linewidth lasers for pumping microresonators allowing the generation of ultra-short pulses and providing ultra-low-noise RF spectral characteristics. But, as we stated above, this platform has much more complicated principles of operation.

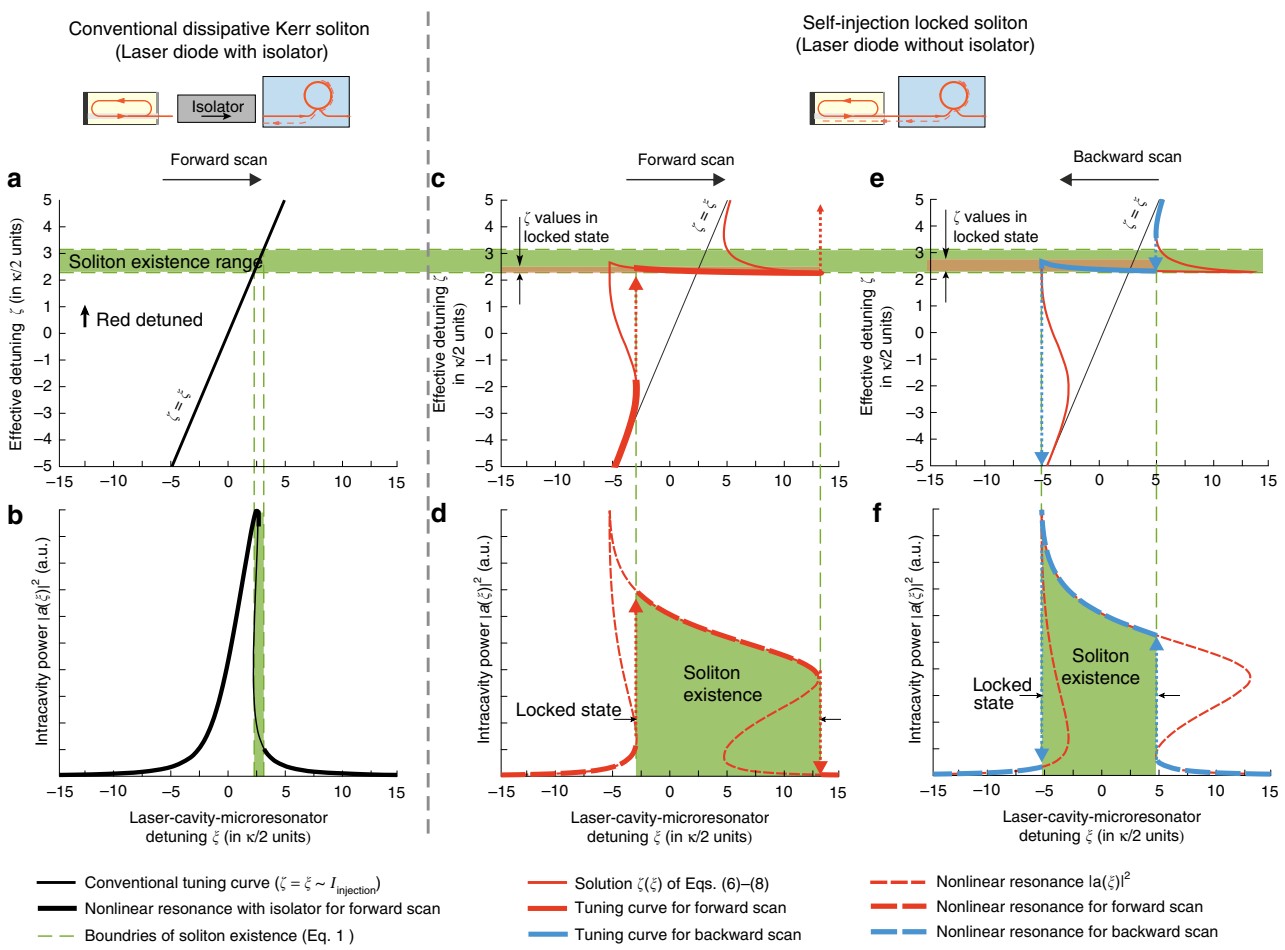

**Fig. 2 Theoretical model of the nonlinear self-injection locking.** Model parameters for **a**–**f**: the normalized pump amplitude $f = 1.6$, the normalized mode-coupling parameter $\Gamma = 0.11$, the self-injection locking coefficient $K_O = 44$, the locking phase $\psi_O = 0.1\pi$. **a**, **b** Model for the conventional case where the microresonator is pumped by the laser with an optical isolator, and $\zeta = \xi$ (dark green line). **c**–**f** Model for the self-injection locked regime. The solution of Eq. (6)–(8) (thin red curves in **c**, **e** is compared with the linear tuning curve $\zeta = \xi$ (thin black lines in **c**, **e**). While tuning the laser, the actual effective detuning $\zeta$ and the intracavity power $|a(\xi)|^2$ will follow red or blue lines with jumps due to the multistability of the tuning curve. The triangular nonlinear resonance curve (thick black in **b**) is deformed when translated from $\zeta$ frame to the detuning $\xi$ frame (**d**, **f**) with corresponding tuning curve $\zeta(\xi)$ (**c**, **e**). The width of the locked state is larger for forward scan, but the backward scan can provide larger detuning $\zeta$, which is crucial for the soliton generation.

To clarify these principles and corroborate our findings from the theoretical model of nonlinear SIL, we develop a technique to experimentally investigate the soliton dynamics via laser SIL, based on a spectrogram measurement of the beatnote signal between the SIL laser and the reference laser. This approach allows the experimental characterization of the nonlinear tuning curve $\zeta(\xi)$, which can be compared to the theoretical model.

First, we measure the microresonator transmission trace by applying 30 Hz triangle diode current modulation from 372 to 392 mA, such that the laser scans over a nonlinear microresonator resonance. As shown in Fig. 4a, the resonance shape is prominently different from the typical triangle shape with soliton steps, characteristic for the conventional generation method, which uses a laser with an isolator, and tunes the laser from the blue-detuned to the red-detuned state (i.e., forward tuning)[45]. However, in the case of the nonlinear SIL, soliton states are observed for both tuning directions, as illustrated below (see Fig. 4) and as it was predicted by the developed model.

Then we measure the tuning curve inside this transmission trace. The reference laser's frequency is set higher than the free-running DFB laser frequency, such that the heterodyne beatnote signal is observed near 15 GHz. The laser diode current is swept rather slow at 10 mHz rate, such that the laser scans across the

resonance in both directions. The spectrogram data in the range of 0 to 25 GHz is collected by ESA (Fig. 4b), and the soliton beatnote signal is measured at 30.696 GHz (Fig. 4c).

Initially, the DFB laser is free-running (Frame I). The diode current and $\xi$ is decreasing, the laser is tuned into a nonlinear microresonator resonance (Frame II). Further decreasing of the diode current locks $\zeta$ to the microresonator inside the soliton existence range (Frame III). In this case, two effects can be observed: the appearance of the 20 GHz beatnote signal (Frame IV) between the reference laser and the first comb line, and the narrow-linewidth and powerful signal of the 30.6 GHz soliton repetition rate, as shown in Fig. 4c (Frame X). The spectrogram regions IV and VII are locally enhanced for better visual representation. Further reducing of the diode current can lead to the switching of (multi-)soliton states. Such switching is caused by the decrease of the effective pump detuning $\zeta$[53], that was predicted by the nonlinear SIL model and is observed in the experiment (see Fig. 4d).

In our experiment, the DFB laser remains locked when the current scan direction reverses from backward to forward (Frame V). This region V, which represents the multi-frequency regime of the DFB (see Supplementary Note 1 for details), is truncated for better visual representation.

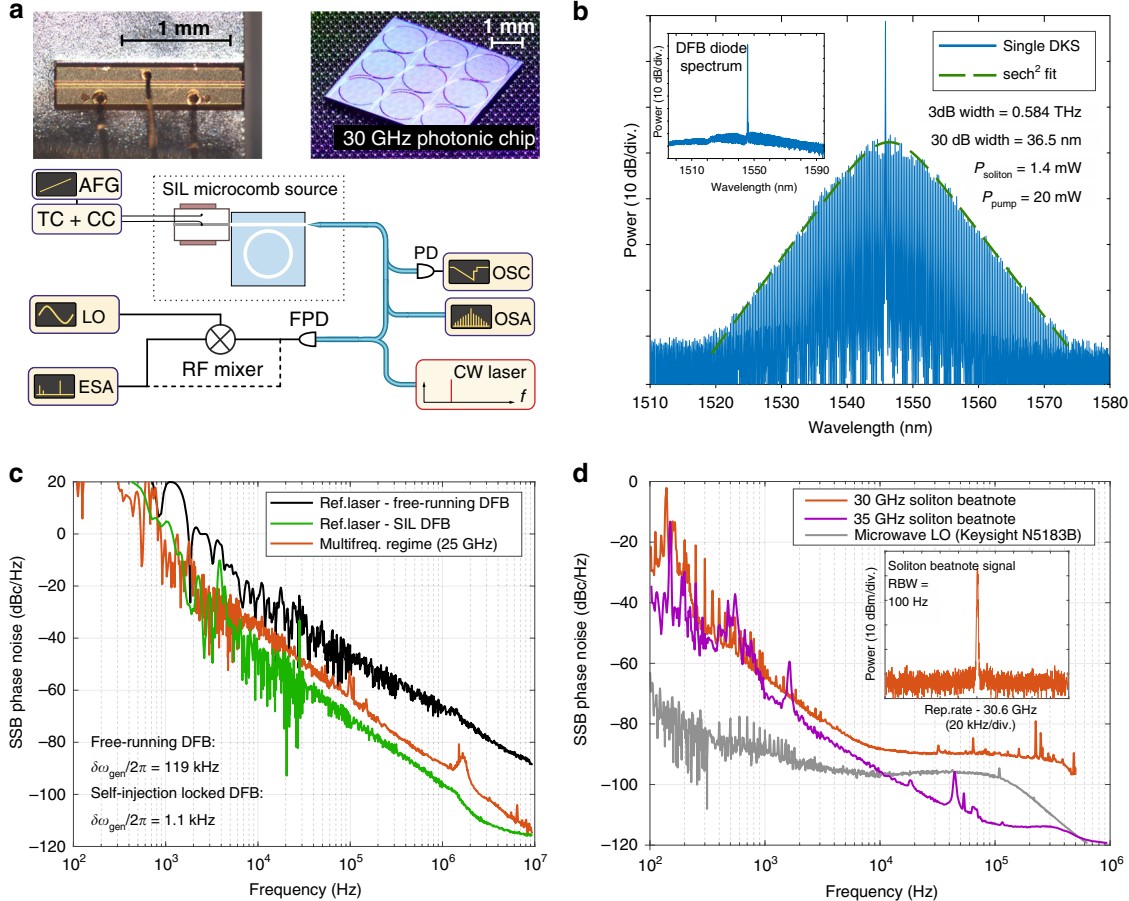

**Fig. 3 Spectral characterization of self-injection locked soliton microcombs. a** Photographs of the DFB laser diode and the Si₃N₄ photonic chip containing 9 microresonators of 30.6 GHz FSR. Experimental setup: the RF mixer is utilized to study RF signal above 26 GHz in combination with a local oscillator (LO). TC: temperature controller. CC: current controller. AFG: arbitrary function generator. OSA: optical spectral analyzer. ESA: electrical spectral analyzer. FPD: fast photodetector. **b** The optical spectrum of the laser self-injection locked single-soliton microcomb. Inset: optical spectrum of the free-running DFB. **c** Comparison of laser phase noise in the free-running and SIL regimes. The phase noise is measured by heterodyning of the system output and CW Laser. The Lorentzian linewidth of the free-running diode is 119 kHz, for the SIL DFB—1.1 kHz. Also, the phase noise in multi-frequency regime of the DFB laser diode is presented (see Supplementary Note 1 for details). **d** Phase noise of the soliton repetition rate signal in different SIL regimes and of LO. The phase noises of 35.4 GHz repetition rate signal are measured directly by high-RF electronics, without RF mixer. Inset: soliton repetition rate signal.

The DFB laser then changes its regime to a single-frequency locked emission again (Frame VI), and generates a chaotic Kerr comb with a wide beatnote signal (Frame XI). As the effective detuning $\zeta$ rises with increasing $\xi$, switching between different soliton states and the first comb line beatnote (Frame VII) is observed, as well as the appearance of the breather soliton states (Frame VIII). Note that the soliton repetition rate reduces with increasing $\zeta$ (Frame X), and vice versa (Frame XII). Further diode current tuning causes the DFB laser to jump out of the SIL regime, its frequency returns to the free-running regime, and thus $\zeta = \xi$ again (Frame IX).

Based on the spectrogram data shown in Fig. 4b, the nonlinear tuning curve $\zeta(\xi)$ is easily extracted, as shown in Fig. 4d, e. These experimentally measured nonlinear tuning curves are fitted using Eq. (6)–(8). In our experiments DFB diode provides $7 < f < 15$. The fitted optical phase is $\psi_0 = -0.27\pi$, and the fitted normalized pump is $f = 13.1$. One may see nearly excellent agreement of theoretical and experimental results. For example, transitions to and from the locked state (Frames II and IX) is perfectly fitted with loops of the theoretical curves in Fig. 4d, e.

Thus, we can conclude that conducted experiments confirm the presence of the non-trivial soliton formation dynamics predicted by the theoretical model. We demonstrate several predicted

effects: first, soliton generation at both directions of the diode current sweep; second, switching of soliton states when the effective detuning $\zeta$ is decreasing[53]. Third, we observe the effect of repetition rate decrease with the growth of the effective detuning $\zeta$ (reported in ref. [57,61]). Measured nonlinear tuning curves are in good agreement with theoretical curves.

## Discussion

In summary, we have studied theoretically and experimentally the effect of the soliton generation by the diode laser self-injection locked to an integrated microresonator. We have developed a theoretical model to describe SIL to a nonlinear microresonator ("nonlinear SIL") and have shown that the complicated dependence of the emission frequency on the injection current leads to the non-trivial dynamics of nonlinear processes. It has been shown that the effective emission frequency of the self-injection locked laser is red-detuned relative to the microresonator resonance and is located inside the soliton existence region for most combinations of parameters. We have checked theoretical results experimentally and we have demonstrated single-soliton generation enabled by a DFB laser self-injection locked to an integrated Si₃N₄ microresonator of 30 GHz

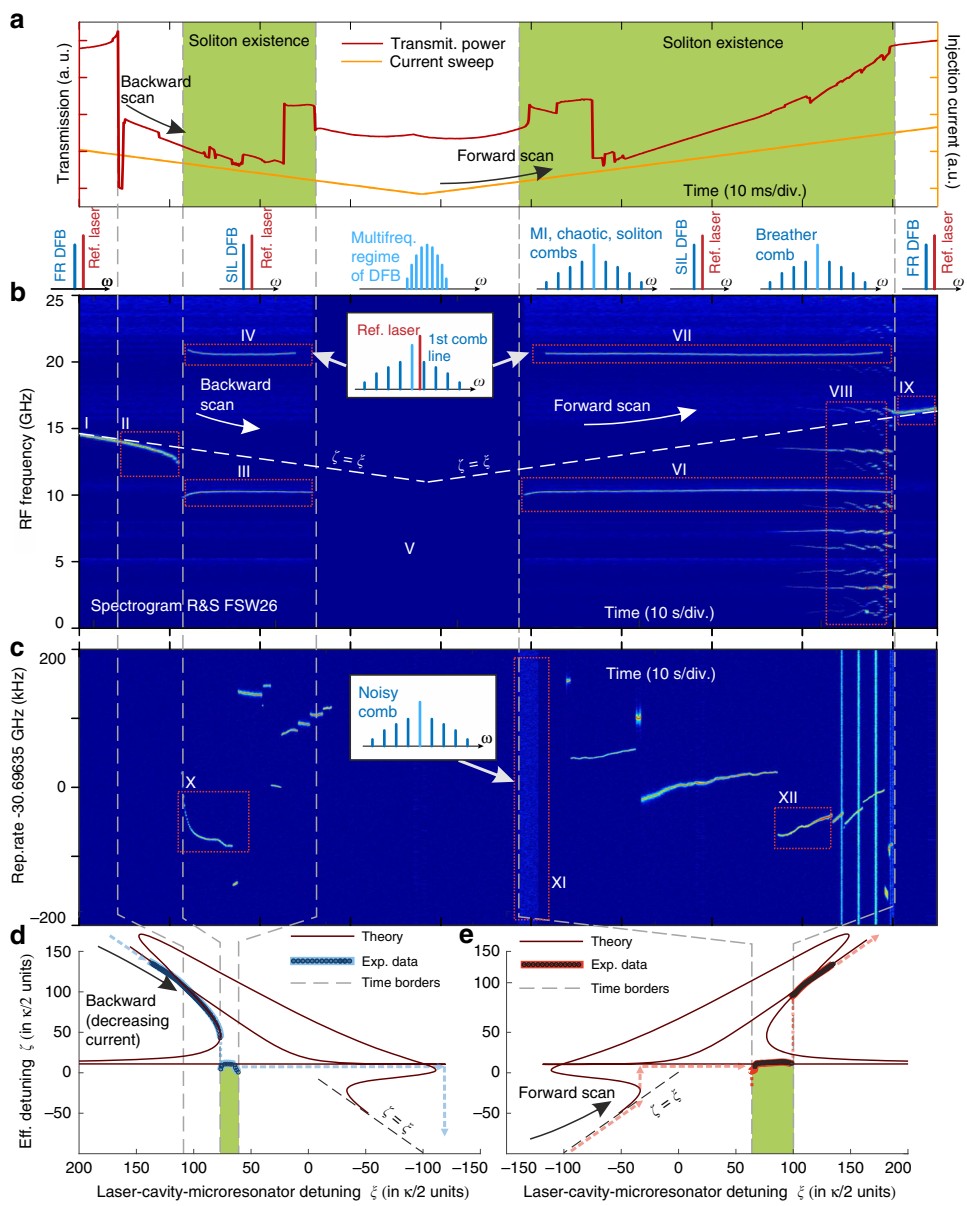

**Fig. 4 Soliton formation dynamics with nonlinear laser self-injection locking. a** Microresonator transmission trace in backward and forward scans, at 30 Hz scan rate. **b** Spectrogram of the beatnote signal between the DFB and the reference laser. **c** Evolution of the soliton repetition rate around 30.6G Hz. **d, e** Measured nonlinear $\zeta(\xi)$ tuning curve and the theoretical fit. The model parameters: normalized pump amplitude $f = 13.1$, the locking phase $\psi_0 = -0.27\pi$ and the locking coefficient $K_0 = 1464$. **a–d** The gray dashed lines correspond to the time borders of different regimes in backward and forward scans. Frames with Roman numerals show different regions of the spectrogram and correspond to the different states of the laser and of the soliton comb: I—free-running laser diode; II—laser diode is locked to the microresonator but the locking is weak and the Kerr comb does not form; III, VI—soliton self-injection locking; IV, VII—the beatnote signal of the 1st comb line and Ref. laser; V—multi-frequency region of laser diode operation (manually truncated); VIII—breather soliton state; IX—free-running laser; X, XII—the soliton rep. rate beatnote signal; XI—chaotic comb.

and 35 GHz FSRs. Also, it has been predicted that in contrast with the free-running laser, in the SIL regime soliton generation is possible for both forward and backward scans of the laser diode injection current. We have developed a spectrogram technique that allows to measure accessed soliton detuning range, and to observe features of nonlinear SIL. Obtained experimental results are in good agreement with theoretical predictions.

Some deviations from the predicted behavior can be attributed to the nonlinear generation of sidebands not included in our theory, which depletes the power in the pumped mode and changes the nonlinear detuning shift (7). Also, the pump power slightly depends on the injection current. Nevertheless, many important theory-derived conclusions have been observed experimentally. Some more important features, predicted by the theory, are yet to study. For example, there are regions of tuning curves with $d\zeta/d\xi = 0$, where noise characteristics of the stabilized laser may be significantly improved.

The radiophotonic signal of the soliton repetition rate provides better phase fluctuations at high offsets (>10 kHz) than the commercial microwave analog signal generator (Keysight N5183B). Note that different soliton states correspond to different phase fluctuations, i.e., optimization of the SIL soliton state may lead to the further decreasing of the phase noise.

Therefore, the soliton SIL provides, first, laser diode stabilization, second, microcomb generation, third, ultra-low noise photonic microwave generation. The main problem of this technique is the limitation of achievable effective detunings: a single-soliton state with a large detuning and broad bandwidth may be hard to obtain in the SIL regime. Further careful parameter optimization is needed for the comb bandwidth enhancement. One possible solution may be based on the fact that backscattering plays a major role and different schemes with increased backscattering may extend the range of effective detunings in the locked state.

Another important question is the increase in the total comb power (see Supplementary Note 7). First, the optimization of the laser diode mode and the photonic chip bus waveguide mode matching is an essential task to decrease the total insertion loss of 7 dB and increase the pump power. Optimization of the parameters of the photonic chip waveguide, particularly speaking, the second-order dispersion $D_2$ and the comb coupling rate to the bus waveguide will lead to the better generation and extraction of the comb lines out of the microresonator[10]. Moreover, bright dissipative Kerr solitons are well-studied structures exhibiting fundamental limitation of the pump-to-comb conversion efficiency. Utilization of the dark soliton pulses, which formation is possible in microresonators with normal GVD, provides Kerr microcombs with high power per each line[62,63]. Moreover, our recent numerical studies suggest that the SIL allows solitonic pulse generation in the normal GVD regime without any additional efforts like mode interaction or pump modulation[64].

Our results provide insight into the soliton formation dynamics via laser SIL, which has received wide interest recently from the fields of integrated photonics, microresonators, and frequency metrology. Also, they may be used for the determination of the optimal regimes of the soliton generation and for the efficiency enhancement of integrated microcomb devices. We believe, that our findings, in combination with recent demonstrations of industrial packaging of DFB lasers and $Si_3N_4$ waveguides, pave the way for highly compact microcomb devices operated at microwave repetition rates, built on commercially available CMOS-compatible components and amenable to integration. This device is a promising candidate for high-volume applications in data-centers, as scientific instrumentation, and even as wearable technology in healthcare. This result is significant for laser systems with strong optical feedback (such as low-noise III-V/Si hybrid lasers and mode-locked lasers), oscillator synchronization, and other laser systems beyond integrated microcombs. A related example microrings are made of quantum cascade active media[65]. Therefore, our findings are relevant not only for integrated photonics community but for a wide range of specialists.

## Methods

**Silicon nitride chip information**. The $Si_3N_4$ integrated microresonator chips were fabricated using the photonic Damascene process[54,66]. The pumped microresonator resonance is measured and fitted including backscattering[50] to obtain the intrinsic loss $\kappa_0/2\pi = 20.7$ MHz, the external coupling rate $\kappa_{ex}/2\pi = 48.6$ MHz, and the backward-coupling rate $\gamma/2\pi = 11.8$ MHz (mode splitting). These correspond to the full resonance linewidth $\kappa/2\pi = \kappa_0/2\pi + \kappa_{ex}/2\pi = 69.3$ MHz, the pump coupling efficiency $\eta = \kappa_{ex}/\kappa = 0.70$, the normalized mode-coupling parameter $\Gamma = \gamma/\kappa = 0.17$, and the amplitude resonant reflection coefficient from the passive microresonator $r \approx 2\eta\Gamma/(1 + \Gamma^2) = 0.23$.

**Butt coupling**. We do not use any optical wire bonding techniques in our work. The DFB laser diode and the $Si_3N_4$ chip are directly butt coupled and are mounted on precise optomechanical stages. The distance between the diode and the chip can be varied with an accuracy better than 100 nm, thus enabling the control of the accumulated optical phase from the $Si_3N_4$ microresonator to the diode (i.e., the locking phase). The output light from the $Si_3N_4$ chip is collected using a lensed fiber. The total insertion loss, i.e., the output free space power of the free-running

laser diode divided by the collected power in the lensed fiber, is 7 dB. Note that matching the polarization of the DFB laser radiation and the polarization of the microcavity high-Q modes is critically important to achieve maximum pump power either via rotation of laser diode or by "polarization engineering" of high-Q modes.

**Experimental setup**. The DFB laser diode' temperature and injection current are controlled by an SRS LD501 controller and an external function generator (Tektronics AFG3102C). The output optical signal of the soliton microcomb is divided by optical fiber couplers and sent to an optical spectrum analyzer (Yokogawa AQ6370D), a fast photodetector (NewFocus 1014), an oscilloscope, and an electrical signal analyzer (ESA Rohde&Schwarz FSW26). The heterodyne beatnote measurement of various comb lines is implemented with a narrow-linewidth reference laser (IDPhotonics DX-2 or TOptica CTL). A passive double-balanced MMIC radio-frequency (RF) mixer (Marki MM1-1140H) is utilized to downconvert and study the RF signal above 26 GHz in combination with a local oscillator (Keysight N5183B). The phase noises of 35.4 GHz repetition rate signal are measured directly by high-RF electronics, without RF mixer.

**Comb generation in SIL regime**. Besides meeting the soliton power budget, a key requirement for soliton generation in the SIL regime is to reach sufficient detuning $\zeta = 2(\omega_0 - \omega_{eff})/\kappa$. When the laser is tuned into resonance from the red-detuned side, SIL can occur so that the laser frequency $\omega_{eff}$ becomes different from $\omega_{LC}$ and is close to $\omega_0$. In the conventional case where soliton initiation and switching are achieved using frequency tunable lasers with isolators, the soliton switching, e.g., from a multi-soliton state to a single-soliton state, can lead to the intracavity power drop which causes the resonance frequency $\omega_0$ shift due to the thermal effect, and ultimately the annihilation of solitons. However, in the case of laser SIL, the optical feedback via Rayleigh backscattering is much faster (instantaneous) than the thermal relaxation time (at millisecond order), therefore the laser frequency can follow the resonance shift instantaneously such that the soliton state is maintained. The slope of the tuning curve $d\zeta/d\xi$ in the SIL regime allows to control the effective detuning $\zeta$ by varying $\xi$, realized by increasing the laser injection current for forward tuning[45], or decreasing the current for backward tuning[53]. However, note that the entire soliton existence range may not be fully accessible at certain locking phases $\psi_0$ and locking coefficient $K_0$. Therefore, a single soliton state with a large detuning and broad bandwidth may be hard to obtain in the SIL regime and further careful parameter optimization is needed. In our experiments, the subsequent switching from chaotic combs to breather solitons and multi-solitons in forward and backward scans is observed (see Supplementary Note 4).

## Data availability
The code and data used to produce the plots within this paper are available at 10.5281/zenodo.4079515. All other data used in this study are available from the corresponding authors upon reasonable request.

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

## Acknowledgements

The authors thank Miles H. Anderson, Joseph Briggs, and Vitaly V. Vassiliev for the fruitful discussion. The Si₃N₄ microresonator samples were fabricated in the EPFL center of MicroNanoTechnology (CMi). DFB diode characterization was made at VNIIOFI center of the shared use (VNIIOFI CSU). This work was supported by the Russian Science Foundation (grant 17-12-01413-П), Contract HR0011-15-C-0055 (DODOS) from the Defense Advanced Research Projects Agency (DARPA), Microsystems Technology Office (MTO), by the Air Force Office of Scientific Research, Air Force Materiel Command, USAF under Award No. FA9550-19-1-0250, and by the Swiss National Science Foundation under grant agreement No. 176563 (BRIDGE) and NCCR-QSIT grant agreement No. 51NF40-185902.

## Author contributions

A.S.V., G.V.L., N.Y.D. conducted the experiment. W.W. conducted the FROG experiment. N.M.K.,V.E.L. developed a theoretical model and performed numerical simulations. J.L. designed and fabricated the Si₃N₄ chip devices. All authors analyzed the data and prepared the manuscript. I.A.B. and T.J.K initiated the collaboration and supervised the project.

## Competing interests

The authors declare no competing interests.
