## [Peer Review File · Nature Communications]

Reviewers' comments:

Reviewer #1 (Remarks to the Author):

Comment on the paper "Dynamics of soliton self-injection locking in optical microresonators, » by Andrey S. Voloshin et al.

This paper reports a theoretical and experimental investigation of soliton comb dynamics in a nonlinear integrated Si₃N₄ microresonator chip pumped by a self-injection locked diode laser. It is shown in particular that the effective emission frequency of the self-injection locked laser is red-detuned relative to the microresonator resonance due to the microresonator's Kerr nonlinearity. It is further shown that the self- and cross-phase modulation of the forward and backward light circulating in the microresonator enables soliton formation with accessible large detunings that are unreachable according to linear self-injection locking theory.

The paper is well written and provides high quality experimental results supported by simulations, although somewhat unclear and lacking in important details. The abstract is relevant, physically speaking. While there is room for improvement and providing more details, this is a rather good result with importance scientifically that may improve our understanding of soliton dynamics in nonlinear chip-based optical resonator. However, I think the paper is not amenable for Nature Communications in its current form and with substantiate claims and thorough completion of all items on the checklist. My main reservation about the paper is that, despite the claim to have generated soliton, there is no plot of temporal soliton pulse including shape and width measurement, nor even any autocorrelation or FROG trace, which would show clear evidence of soliton related to comb spectrum shown in Fig. 3b. This is a serious omission and in my view makes the paper less interesting and significant.

Reviewer #2 (Remarks to the Author):

The manuscript provides a detailed analysis of soliton self-injection locking, i.e. that the feedback of the microresonator in the pumping laser will affect the laser frequency, which in turn will influence the nonlinear dynamics of the microresonator. This can lead to the stabilization of a frequency comb relaxing needs on other resource demanding and complex active and passive stabilization schemes, or to a further destabilization of the frequency comb, if things are done wrongly. The problem is demanding as it evolves the coupled dynamics of two systems displaying already complex nonlinear dynamics. Due this challenge and the potential impact on the communities in metrology, integrated photonics and nonlinear dynamics I welcome the publication in a high profile journal. The investigation and results seem to be also sound and well done, although I have a few questions on the model. Most figures (in particular Fig. 2) are also done nicely to illustrate the complex relationships. Hence I recommend publication, if the authors clarify the following points.

1. Section II: In the first paragraph ω_0 is introduced without a proper definition. In the second paragraph ω_0 is introduced as the "loaded" cavity resonance. I do not understand this phrasing as the model is linear. It should be explained explicitly, what is included in ω_0 and what not. In addition, no model is presented on the locking behaviour of the lasers diode. This should include the Henry factor, introduced only much later. I assume the corresponding equations are already used here. This needs to be rewritten.
2. The literature list is very heavy on the papers of the authors. In particular, there is a long tradition of papers on laser diode feedback and self-injection locking, which should be acknowledged. E.g. Lang IEEE JQE 16, 347 (1980), Petermann IEEE Selec Top QE 1, 480 (1995); Zorabedian IEEE QE 23, 1855 (1987), Tkach IEEE JLWT 4, 1655 (1986). The latter also classifies feedback regimes. Is the notion of regime V in the supplementary material linked to the classification or is it an own one? The equation for the self-injection coefficient after Eq. (6) would probably justify an original citation.
3. Fig. 1c could be easier to understand: The most important curve is actually the red one but it is the

thinnest. It is completely hidden by the thick light red dashes indicating the locking region in that region. Should the dashes extend beyond the locking regime? Probably not. Similar, but not quite so pronounced in Fig. 1d.

4. Fig. 4d is quite unclear and not explained well. The thin grey lines have no legend. I do not think that the two straight pieces of thick solid black line is the theory as it seems to be indicated in the legend. Similarly, the experimental data need to be better identified.

Reviewer #3 (Remarks to the Author):

The authors present theoretical investigations of self-injection locking of a DFB laser to a silicon nitride microresonator for Kerr comb generation and soliton formation. They investigate the soliton existence region for forward and backward tuning of the laser-cavity detuning. The paper is generally well written and would be of specific interest to the Kerr comb community. However, there has been many prior demonstrations of injection locking for Kerr comb generation and soliton formation in recent years, including the work done by some of the authors of this manuscript, including Pavlov, et al., "Narrow linewidth lasing and soliton Kerr microcombs with ordinary laser diodes," *Nat. Photon.* 12, 694 (2018); Raja, et al., "Electrically pumped photonic integrated soliton microcomb," *Nat. Commun.* 10, 680 (2019); Stern, et al., "Battery-operated integrated frequency comb generator," *Nature* 562, 401 (2018); Raja, et al., "Packaged photonic chip-based soliton microcomb using an ultralow-noise laser," arXiv:1906.03194; Raja, et al., "Chip-based soliton microcomb module using a hybrid semiconductor laser," *Opt. Express* 28, 2714 (2020); Lesko, et al., "Fully phase-stabilized 1 GHz turnkey frequency comb at 1.56 μm ," arXiv:2005.03088; Shen, et al., "Integrated turnkey soliton microcombs operated at CMOS frequencies," arXiv:1911.02636. While the experimental investigation of the noise performance is a nice addition, the overall performance of the comb source is similar to the previous demonstrations by the authors and do not see any notable advance. Especially with many prior experimental work being demonstrated using the self-injection locked scheme, the theoretical work will be more suitable in a specialized journal rather than a journal with broad readership such as *Nature Communications*. Specific comments are below.

1. What is the required strength of the backscattered light for injection locking to occur? What is the value in the experiment?
2. In Fig. 3c, what do the authors mean by Ref. laser -Ref. laser?
3. In Figure 4, can the authors clarify the tuning path for backward scan? How is the resonance initially captured? In general the figure caption is incomplete. The authors should address what the Roman numerals are for each of the regions described in b.
4. The generated comb spectra is rather narrowband. There have been many demonstrations of broadband soliton microcomb generation in silicon nitride microresonators. Is this possible in an injection locked scheme? What, if any, are the limitations for achieving broad bandwidth?

First of all, we would like to express our deep appreciation to the three reviewers, for the time they took to carefully read our manuscript and the concerns that help us improve the quality of our manuscript. We appreciate that the reviewers read it carefully, made suggestions and gave overall positive evaluations:

- *“While there is room for improvement and providing more details, this is a rather good result with importance scientifically that may improve our understanding of soliton dynamics in nonlinear chip-based optical resonator”* (Reviewer #1)
- *“The investigation and results seem to be also sound and well done.”* (Reviewer #2)

In the following, we will respond in great detail (in black) to the reviewers' questions (in blue), point-by-point, as well as the action taken (in red):

Executive Summary:

In the revised manuscript, according to the Reviewer #1's request, we have added new experimental data to the SI – the FROG measurement confirming that the soliton self-injection locking provides generation of pulses in the temporal domain. Also, we measured the ultra-low phase noise of the photonic microwave signal generated by the soliton. We emphasize that the key novelty and scientific significance of our work is the comprehensive study of the dynamics of soliton self-injection locking corroborated by a novel quantitative theoretical model. We perform soliton beatnote spectroscopy and demonstrate real-time switching in the self-injection locking regime between different soliton states which have not been studied before. Our results could be applied to laser systems beyond Kerr microcombs, such as photonic circuits containing both nonlinear photonic elements and active media and microresonators made of active materials.

Reviewer #1

This paper reports a theoretical and experimental investigation of soliton comb dynamics in a nonlinear integrated Si₃N₄ microresonator chip pumped by a self-injection locked diode laser. It is shown in particular that the effective emission frequency of the self-injection locked laser is red-detuned relative to the microresonator resonance due to the microresonator's Kerr nonlinearity. It is further shown that the self- and cross-phase modulation of the forward and backward light circulating in the microresonator enables soliton formation with accessible large detunings that are unreachable according to linear self-injection locking theory.

The paper is well written and provides high quality experimental results supported by simulations, although somewhat unclear and lacking in important details. The abstract is relevant, physically speaking. While there is room for improvement and providing more details, this is a rather good result with importance scientifically that may improve our understanding of soliton dynamics in the nonlinear chip-based optical resonator. However, I think the paper is not amenable for Nature Communications in its current form and with substantiate claims and thorough completion of all items on the checklist. My main reservation about the paper is that, despite the claim to have generated soliton, there is no plot of temporal soliton pulse including shape and width measurement, nor even any autocorrelation or FROG trace, which would show clear evidence of soliton related to the comb spectrum shown in Fig. 3b. This is a serious omission and in my view makes the paper less interesting and significant.

Our reply:

First of all, we thank the reviewer for the positive comments. We also appreciate that the reviewer shares his concern with us on the significance of our study. We clarify that the main focus of

our work is to describe and characterize the dynamics of soliton formation via laser self-injection locking in Kerr nonlinear optical microresonators.

The first FROG measurements of dissipative Kerr solitons in optical microresonators was presented in Ref. [R1, R2]. It has been shown that the FROG measurement in microresonators distinguished a pulsed waveform from a non-pulsed, periodic, modulated waveform. However, FROG is not the only criterion in determining if the comb state is indeed a soliton pulse. It is now widely accepted in the microcomb community that two measurements [R3] could suffice to prove the existence of soliton pulses in microresonators: (1) the sech^2 profile of the optical spectrum (in the case of the single soliton state), and (2) low phase noise of the beatnote of the soliton repetition rate. Self-injection locked solitons meet both criteria. Particularly, in our original manuscript, we have shown the low-phase-noise beatnote signal of the soliton repetition rate, which is convincing evidence of soliton existence and demonstrates the mode-locked regime. For this reason, we did not include the FROG measurement in our initial submission.

To respond to the reviewer's request, and to prove a pulsed waveform, we have performed additional experiment, and have measured the temporal profile of soliton pulses with FROG. The photonic chip-based microresonator used here has an FSR of 35.397 GHz. The data is presented in Fig. F1.

Before the FROG setup, the optical spectrum of self-injection locked soliton is filtered by a fiber Bragg grating (FBG) for pump suppression and amplified to ~100 mW by using two EDFAs. The FBG suppressed not only the central comb line but also neighbouring lines because the FBG's bandwidth (~100GHz) is larger than the comb lines spacing (35 GHz). Also, the optical amplifiers used here have a gain profile which is not flat. This causes the distortion of the soliton's optical spectrum. The reconstructed intracavity power from the FROG trace proves that the spectrum corresponds to a short pulse train. We confirmed the pulsed waveform of the soliton microcomb. A detailed study of soliton parameters is not the key point of our work, so we did not analyze the FROG trace fine structure.

Moreover, we would like to mention that our theoretical model of the self-injection locking is valid for microresonators with normal group velocity dispersion (GVD). In these microresonators, bright solitons do not exist, while dark pulses, i.e. platicons, can be observed [R4]. The principles of laser self-injection locking remain the same in this case of normal-GVD microresonators. Our approach was successfully used for the normal-GVD microresonators, and the preliminary result has been presented at CLEO US 2020 [R5].

Figure F1: Frequency Resolved Optical Gating (FROG) of self-injection locked soliton. (a) Original FROG trace measured for soliton microcomb with a repetition rate of 35.397 GHz is on the left panel. The microcomb was, first, filtered by using fiber Bragg grating to suppress the central line and amplified to ~100 mW by using cascaded optical amplifiers (with non-flat gain). That is why the FROG trace does not correspond to the pure single soliton temporal profile, but still exhibits the optical pulse with a width of less than 1 picosecond. The reconstructed FROG trace is on the right panel. (b) Optical spectrum of 35.397 GHz self-injection locked soliton. (c) The reconstructed intracavity power of the soliton pulse. Deviation and broadening from the expected soliton temporal profile are caused by the changes in the optical spectrum.

[R1] Herr, T., Brasch, V., Jost, J. et al. "Temporal solitons in optical microresonators," *Nature Photon* 8, 145–152 (2014).

[R2] Xu Yi, Qi-Fan Yang, Ki Youl Yang, Myoung-Gyun Suh, and Kerry Vahala, "Soliton frequency comb at microwave rates in a high-Q silica microresonator," *Optica* 2, 1078-1085 (2015)

[R3] Gaeta, A.L., Lipson, M. & Kippenberg, T.J. "Photonic-chip-based frequency combs," *Nature Photon* 13, 158–169 (2019).

[R4] S.-W. Huang, H. Zhou, J. Yang, J. F. McMillan, A. Matsko, M. Yu, D.-L. Kwong, L. Maleki, and C.W. Wong, "Mode-locked ultrashort pulse generation from on-chip normal dispersion microresonators," *Phys. Rev. Lett.* 114, 053901 (2015).

[R5] Grigory V. Lihachev, et al., "Laser Self-Injection Locked Frequency Combs in a Normal GVD Integrated Microresonator," *CLEO Conference 2020, STh10.3* (2020).

Action taken:

- We added the Supplementary Note 6 to the Supplementary Materials, where we explain how one can prove the soliton presence in microresonators, demonstrate original FROG trace, reconstructed FROG trace and pulse waveform.
- We added the following text in the abstract (marked in red) on the 1st page in the revised manuscript:
“...locking theories. We construct state-of-the-art integrated soliton microcombs with an electronically detectable repetition rate of 30 GHz and 35 GHz with as low phase noises as -96 dBc at 10 kHz frequency offset. These devices, consisting of a DFB laser self-injection-locked to a Si₃N₄ microresonator chip, allow us to implement a novel experimental technique and study the soliton formation dynamics as well as the repetition rate evolution in real-time. Conducted experiments...”
- We added the following text to Section IV “Self-injection locking to a nonlinear microresonator” (page 4): “Having access to the stable operation of these soliton states, we prove experimentally (in addition to their ultra-low noise RF beatnote signal (Fig. 3d, inset), optical spectrum (Fig. 3b) and zero background noise) that such optical spectra correspond to the ultra-short pulses representing bright temporal solitons. We perform a frequency-resolved optical gating (FROG) experiment [45]. This corresponds to a second-harmonic generation autocorrelation experiment in which the frequency doubled light is resolved in spectral domain (see Supplementary Note 6 for details). Reconstructed optical field confirms that we study temporal solitons with width less 1 picosecond. Therefore, self-injection locking is a reliable [33, 37] platform to substitute bulky narrow-linewidth lasers for pumping microresonators allowing the generation of ultra-short pulses and providing ultra-low noise RF spectral characteristics. But, as we stated above, this platform has much more complicated principles of operation.”

Reviewer #2

The manuscript provides a detailed analysis of soliton self-injection locking, i.e. that the feedback of the microresonator in the pumping laser will affect the laser frequency, which in turn will influence the nonlinear dynamics of the microresonator. This can lead to the stabilization of a frequency comb relaxing needs on other resource demanding and complex active and passive stabilization schemes, or to a further destabilization of the frequency comb, if things are done wrongly. The problem is demanding as it evolves the coupled dynamics of two systems displaying already complex nonlinear dynamics. Due this challenge and the potential impact on the communities in metrology, integrated photonics and nonlinear dynamics I welcome the publication in a high profile journal. The investigation and results seem to be also sound and well done, although I have a few questions on the model. Most figures (in particular Fig. 2) are also done nicely to illustrate the complex relationships.

1. Section II: In the first paragraph ω_0 is introduced without a proper definition. In the second paragraph ω_0 is introduced as the “loaded” cavity resonance. I do not understand this phrasing as the model is linear. It should be explained explicitly, what is included in ω_0 and what not. In addition, no model is presented on the locking behaviour of the lasers diode. This should include the Henry factor, introduced only much later. I assume the corresponding equations are already used here. This needs to be rewritten.

Our reply:

The word “loaded” in the sentence was intended to refer only to the cavity linewidth, not the frequency. We meant that “ ω_0 is the frequency of the microresonator resonance and κ is its loaded linewidth”. As ω_0 is the frequency of the microresonator, the Henry factor is not related to it. However, it is, of course, included in the laser cavity frequency ω_{LC} as it is by definition a frequency of the free-running laser.

In this section we discuss only the general principle of the self-injection locking. In the following theoretical section we provide more clarifications that ω_μ are cold microresonator resonances. We also indicate where the reader can find the locking behaviour of the laser diode. The derivation of the tuning curve equations is shown and explained in the text. We suppose that the detailed description of the methods of laser and microresonator eigenfrequency calculation will greatly complicate the article.

Action taken:

- We changed the wording to avoid the confusion: «...where ω_0 is the frequency of the microresonator resonance and κ is its loaded linewidth.»
- We add the following sentence to the Section II “Principle of laser self-injection locking”: «We also note that the laser cavity resonant frequency ω_{LC} as well as ξ are also assumed to include the Henry factor in its definition.»
- To clarify the laser model definition, we add the following sentence: “For analysis of the SIL effect we combine the eqs. (2) with the standard laser rate equations similar to the Lang-Kobayashi equations [39], but with resonant feedback [44].”

[39] Lang, R. & Kobayashi, K. IEEE Journal of Quantum Electronics 16, 347–355 (1980).

[44] Kondratiev, N. M. et.al. Opt. Express 25, 28167–28178 (2017).

2. The literature list is very heavy on the papers of the authors. In particular, there is a long tradition of papers on laser diode feedback and self-injection locking, which should be acknowledged. E.g. Lang IEEE JQE 16, 347 (1980), Petermann IEEE Select Top QE 1, 480 (1995); Zorabedian IEEE QE 23, 1855 (1987), Tkach IEEE JLWT 4, 1655 (1986).

The latter also classifies feedback regimes. Is the notion of regime V in the supplementary material linked to the classification or is it an own one? The equation for the self-injection coefficient after Eq. (6) would probably justify an original citation.

Our reply:

We appreciate the reviewer for providing us with these classical works. All mentioned papers are about frequency-independent non-resonant feedback and their derivations are generally not applicable to the highly selective backscattering of the high-Q microresonator. We added references to the papers in several places in the revised manuscript.

To answer “The latter also classifies feedback regimes. Is the notion of regime V in the supplementary material linked to the classification or is it an own one”: If we understand the question correctly, we are speaking about the regimes of the diode emission, according to Fig. 4. The regime of multi-frequency (regime V on the spectrogram) is characterized by the rise of suppressed laser diode modes, being suppressed in the normal regime. That is why the beatnote signal at the frequency corresponding to the laser diode cavity length is observed. We do not suppose that this regime is similar to one described in Ref. [52], because the laser emission frequency remains narrow and locked to the microresonator. Tkach's classification does not work for the narrow-band feedback systems. One can argue that the high-Q cavity can be viewed as a long arm with effective distance derived from the Q-factor thus making the dynamics of such system to be inside the Tkach's V regime, but this seems to be unnecessarily complicated. If we refer to the reviewer question to SUPPLEMENTARY NOTE 2 and Fig.3 in the SI, then the different regimes correspond to different

optical phases and do not correspond to the regimes from [52] and are just five exemplary traces without any classification purposes.

To answer “The literature list is very heavy on the papers of the authors”: we’ve deleted some citations of the authors.

Action taken:

- The last paragraph of the introduction was modified as follows:
“However, despite the inspiring and promising experimental results the principles and dynamics of the soliton self-injection locking have never been thoroughly studied. Only recently some aspects of the soliton generation effect were investigated [37], where a static operation was considered, but a comprehensive theoretical and experimental investigation is still necessary. The common SIL models consider either laser equations with frequency-independent feedback [39-41], or linear-resonant feedback [42-44]. Here, we first develop an original theoretical model, taking into account nonlinear interactions of the counter-propagating waves in the microresonator, to describe nonlinear SIL, i.e. SIL to a nonlinear microresonator.”
- We add the following sentence to the Section II “Principle of laser self-injection locking”: «We also note that the laser cavity resonant frequency ω_{LC} as well as ξ are also assumed to include the Henry factor in its definition.»
- To clarify the laser model definition, we add the following sentence: “For analysis of the SIL effect we combine Eqs. (2) with the standard laser rate equations similar to the Lang-Kobayashi equations [39], but with resonant feedback [44].”
- We add the following after the equation (6): “The self-injection locking coefficient K_0 is analogous to the feedback parameter C used in the theory of the simple mirror feedback [40, 41], where the self-injection is achieved with the frequency-independent reflector forming an additional Fabry-Perot cavity. However, in the resonant feedback setup, the self-injection locking coefficient does not depend on the laser-to-reflector distance, depending on the parameters of the reflector instead. Though the system has qualitatively similar regimes as a simple one [52], their ranges and thresholds are different [42, 44, 46]. The value of $K_0 > 4$ is required for pronounced locking with the sharp transition, naturally becoming a locking criterion. For high-Q microresonators, this value can be no less than several thousand. We also note that in linear regime (or in nonlinearly shifted coordinates $\underline{\xi}$, $\underline{\zeta}$) the stabilization coefficient of the setup is close to K_0 , full locking range is close to $0.65K_0$.”
- Also, we have deleted the following citations of authors: <https://doi.org/10.1109/CLEOE-EQEC.2019.8873388>, <https://doi.org/10.1051/epjconf/201922002006>, <http://www.osapublishing.org/abstract.cfm?URI=ASSL-2019-JTu3A.32>

[52] Tkach, R. & Chraplyvy, A. *Journal of Lightwave Technology* 4, 1655–1661 (1986).

[37] Shen, B et.al. *Nature* 582, 365–369 (2020).

[39] Lang, R. & Kobayashi, K. *IEEE Journal of Quantum Electronics* 16, 347–355 (1980).

[40] Petermann, K. *IEEE Journal of Selected Topics in Quantum Electronics* 1, 480–489 (1995).

[41] Donati, S. *Laser & Photonics Reviews* 6, 393–417 (2012).

[42] Laurent, P. et.al. *IEEE Journ. Quant. El.* 25, 1131–1142 (1989).

[43] R. Kazarinov and C. Henry, *IEEE Journal of Quan. Elect.*, vol. 23, no. 9, pp. 1401-1409, (1987).

[44] Kondratiev, N. M. et.al. *Opt. Express* 25, 28167–28178 (2017).

[46] Galiev, R. R., et al. *Phys. Rev. Applied* 14, 014036 (2020).

3. Fig. 1c could be easier to understand: The most important curve is actually the red one but it is the thinnest. It is completely hidden by the thick light red dashes indicating the locking region in

that region. Should the dashes extend beyond the locking regime? Probably not. Similar, but not quite so pronounced in Fig. 1d.

Our reply:

We thank the reviewer for the suggestion to make this important figure better.

To answer “Should the dashes extend beyond the locking regime?”: We would like to highlight the range of ζ in the locked state in the revised manuscript.

Action taken:

- We revised the Fig. 1 according to the reviewer’s comment. We changed the dashed region so that it corresponds to the locking regime only.

4. Fig. 4d is quite unclear and not explained well. The thin grey lines have no legend. I do not think that the two straight pieces of thick solid black line is the theory as it seems to be indicated in the legend. Similarly, the experimental data need to be better identified.

Our reply:

We thank the reviewer for the suggestion. There were some graphical artifacts which have been deleted in our revised manuscript. Also, we improved the visual representation of the experimental data. We added the legend for thin grey lines.

Action taken:

- We revised the Fig. 4d.
-

Reviewer #3

The authors present theoretical investigations of self-injection locking of a DFB laser to a silicon nitride micresonator for Kerr comb generation and soliton formation. They investigate the soliton existence region for forward and backward tuning of the laser-cavity detuning. The paper is generally well written and would be of specific interest to the Kerr comb community.

However, there has been many prior demonstrations of injection locking for Kerr comb generation and soliton formation in recent years, including the work done by the some of the authors of this manuscript, including Pavlov, et al., “Narrow-linewidth lasing and soliton Kerr microcombs with ordinary laser diodes,” Nat. Photon. 12, 694 (2018); Raja, et al., “Electrically pumped photonic integrated soliton microcomb,” Nat. Commun. 10, 680 (2019); Stern, et al., “Battery-operated integrated frequency comb generator,” Nature 562, 401 (2018); Raja, et al., “Packaged photonic chip-based soliton microcomb using an ultralow-noise laser,” arXiv:1906.03194; Raja, et al., “Chip-based soliton microcomb module using a hybrid semiconductor laser,” Opt. Express 28, 2714 (2020); Lesko, et al., “Fully phase-stabilized 1 GHz turnkey frequency comb at 1.56 μm ,” arXiv:2005.03088; Shen, et al., “Integrated turnkey soliton microcombs operated at CMOS frequencies,” arXiv:1911.02636.

While the experimental investigation of the noise performance is a nice addition, the overall performance of the comb source is similar to the previous demonstrations by the authors and do not see any notable advance. Especially with many prior experimental work being demonstrated using the self-injection locked scheme, the theoretical work will be more suitable in a specialized journal rather than a journal with broad readership such as Nature Communications.

Our reply:

First of all, we thank the reviewer for reviewing our work and share the comments. We kindly but firmly disagree with the statement that in our work “the overall performance of the comb source is similar to the previous demonstrations by the authors and do not see any notable advance.” First, we would like to explain and highlight the significance of our work, compared with previous works mentioned by the reviewer:

1. Our work is the first report presenting the study of the soliton self-injection locking dynamics via theoretical model, numerical simulations and experimental characterization. Prior works, including Ref. [R1-R3], studied only static comb states and only mentioned that switching of the soliton combs is possible. We simulate, for the first time, the self-injection locking dynamics taking into account the microresonator’s Kerr nonlinearity, so-called “nonlinear laser self-injection locking”. These results will become the point of interest not only for the Kerr comb community but for a wide community of photonics researchers because the integration of active structures with passive photonic integrated circuits and production of micro-rings from active material (for example, QCL) is in great demand.
2. We perform soliton beatnote spectroscopy and demonstrate switching in the self-injection locking regime between different soliton states, which has not been studied before. The measurement of the laser emission frequency depending on the laser injection current (Fig. 4 in the main text) has not been performed in previous works despite its importance to explain the locking dynamics. There were only attempts to measure such curves (Ref. [R4]). The researchers mistakenly expected that frequency tuning curves in the nonlinear case were similar to linear ones.
3. In this sense, our manuscript should be compared with Ref. [R5], because in both works the switching of soliton combs is investigated. Reference [R5] studied the conventional method of soliton initiation in the presence of an optical isolator (which is not amenable for photonic integration), while our work studies the case of fully integrated microcombs operating with self-injection locking (in the absence of an optical isolator).
4. Our work [R6] was done concurrently and independently with the paper Shen, et al., “Integrated turnkey soliton microcombs operated at CMOS frequencies,” arXiv:1911.02636 (placed on arXiv back to back in November and December, respectively). Ref. [R5] presents a key step in photonic integration and packaging of DFB lasers to Si₃N₄ photonic chips, but it contains only a qualitative analysis of laser self-injection locking in the static model in contrast with ours. That paper has been published in Nature recently.

Also, we strongly disagree that our work is “the theoretical work” and “will be more suitable in a specialized journal rather than a journal with broad readership such as Nature Communications.” Our work includes thorough experimental results explained by the proposed novel theoretical model. Understanding of the soliton self-injection locking helps us to generate the single-soliton state with the lowest repetition rate of 30 GHz, that is the record to our knowledge. The community quickly adopted our approach to investigate self-injection locking. Please, let me show how our work is described in recent papers:

- “*Further theoretical investigation of the self-injection dynamics based on an adapted model like the one proposed by A. S. Voloshin and coworkers [R6] may explain the behavior of our system.*” - the published paper [R7] of Sylvain Boust (the group of Frédéric van Dijk, III-V lab in Thales Research and Technology, France).
- “*However, a recent theoretical and experimental demonstration of self-injection locking shows that the dynamics of self-injection locking is sensitive to various parameters, e.g., strength and phase of the backscattering and pump power [R6]*” - preprint [R8].
- “*So here’s one of these tuning curves for $F^2 = 10$, and again this is based on a very nice paper [R6]*” - CLEO 2020 presentation of Travis Briles (the group of Scott Papp in NIST) [R9]. Our approach was chosen by NIST to investigate the integrated octave-spanning frequency comb, which they developed.

We have to note that Ref. [R10, R11] study breadboard implementation of Kerr soliton combs, i.e. the microresonator is pumped by external laser. And Ref. [R12] utilizes portable but not integrated frequency comb based on a fiber mode-locked laser.

- [R1] Pavlov, et al., "Narrow-linewidth lasing and soliton Kerr microcombs with ordinary laser diodes," Nat. Photon. 12, 694 (2018)
- [R2] Raja, et al., "Electrically pumped photonic integrated soliton microcomb," Nat. Commun. 10, 680 (2019)
- [R3] Stern, et al., "Battery-operated integrated frequency comb generator," Nature 562, 401 (2018);
- [R4] Savchenkov, Anatoliy, Skip Williams, and Andrey Matsko. "On stiffness of optical self-injection locking." Photonics. Vol. 5. No. 4. Multidisciplinary Digital Publishing Institute, 2018.
- [R5] Guo, H., Karpov, M., Lucas, E. et al. Universal dynamics and deterministic switching of dissipative Kerr solitons in optical microresonators. Nature Phys 13, 94–102 (2017).
- [R6] our manuscript - <https://arxiv.org/pdf/1912.11303>
- [R7] S. Boust et al., "Microcomb source based on InP DFB / Si₃N₄ microring butt-coupling," in Journal of Lightwave Technology, doi: 10.1109/JLT.2020.3002272.
- [R8] K. Nishimoto, K. Minoshima, T. Yasui, and N. Kuse, "Generation of a microresonator soliton comb pumped by a DFB laser with phase noise measurements," <https://arxiv.org/abs/2002.00736> (2020)
- [R9] T. C. Briles et al., "Semiconductor laser integration for octave-span Kerr-soliton frequency combs," CLEO Conference 2020, STh1O.6, 2020
- [R10] Raja, et al., "Packaged photonic chip-based soliton microcomb using an ultralow-noise laser," arXiv:1906.03194
- [R11] Raja, et al., "Chip-based soliton microcomb module using a hybrid semiconductor laser," Opt. Express 28, 2714 (2020)
- [R12] Lesko, et al., "Fully phase-stabilized 1 GHz turnkey frequency comb at 1.56 μm," arXiv:2005.03088

1. What is the required strength of the backscattered light for injection locking to occur? What is the value in the experiment?

Our reply:

The strength of backscattering can be defined in several ways. First, it may mean the coupling rate described by the normalized mode-coupling parameter Γ (see Si₃N₄ chip information in Methods). This parameter can be calculated by fitting the microresonator's resonance and directly measured if the resonance splitting is detectable. It is easy to do because this parameter characterizes the microresonator, not the whole system "diode-microresonator", and one can do it for any microresonator in the external setup by coupling the light into the bus waveguide by the lensed fiber. For values $\Gamma < 0.16$, it approximately equals the amplitude reflection coefficient in resonance. One can measure the backscattered light using beam-splitter or by using the circulator.

We measured $\Gamma = 0.17$ by fitting the resonance profile (see Methods). Also, we measured the resonant backscattered light power to be ~5-10% of input power depending on particular mode by using the circulator.

Usually, a small value of Γ is sufficient to trigger self-injection locking. We saw the self-injection locking for high-Q crystalline microresonators with Γ as low as 0.01.

The linewidth reduction depends on $(Q_{\text{microresonator}}/Q_{\text{diode}})^2$, so the influence of the microresonator's Q-factor is much higher than the influence of the normalized mode-coupling, because $Q_{\text{diode}} \sim 10^4$ and $Q_{\text{microresonator}} > 10^6$. In other words, one can obtain ultranarrow linewidth in the self-injection locking regime even with very low backscattering.

Second, another option to describe quantitatively the strength of the backscattering is the self-injection locking coefficient K_0 , which is similar to the classical feedback parameter C [40]. In some

approximations the linewidth reduction $\frac{\delta\omega_{free}}{\delta\omega_{locked}} \approx K_0^2$ and the locking range $\frac{2\Delta\omega_{lock}}{K} \approx 0.65K_0$.

[40] Petermann, K. IEEE Journal of Selected Topics in Quantum Electronics 1, 480–489 (1995). <http://dx.doi.org/10.1109/2944.401232>

Action taken:

- This issue is clarified in the revised version. We add the following after the equation (6) (page 4):
“The self-injection locking coefficient K_0 is analogous to the feedback parameter C used in the theory of simple mirror feedback [40, 41]... The value of $K_0 > 4$ is required for pronounced locking with the sharp transition, naturally becoming a locking criterion. For high-Q microresonators, this value can be no less than several thousand...”

2. In Fig. 3c, what do the authors mean by Ref. laser -Ref. Laser?

Our reply:

Ref.laser - Ref.laser means the phase noise calculated by the heterodyning of two identical reference lasers (IDPhotonics DX-2).

Action taken:

- The legend is changed to “Ref. laser phase noise”

3. In Figure 4, can the authors clarify the tuning path for backward scan? How is the resonance initially captured? In general the figure caption is incomplete. The authors should address what the Roman numerals are for each of the regions described in b. In general the figure caption is incomplete. The authors should address what the Roman numerals are for each of the regions described in b.

Our reply:

We have measured the nonlinear resonance (the microresonator transmission trace) on the oscilloscope directly. A 30 Hz triangle modulation was applied on the laser diode current from 372 to 392 mA, such that the laser frequency scans over the resonance in backward and then in forward directions. Then the frequency tuning curve inside this diode current range was measured. The reference laser's frequency is set higher than the free-running DFB laser frequency, such that the heterodyne beatnote signal is observed near 15 GHz. Then we slow the triangle modulation down to 10 mHz and observe the heterodyne beatnote signal in the spectrogram regime.

First, the laser diode is free-running (frame I). Then the laser diode is locked to the microresonator but the locking is weak and the Kerr comb does not form (frame II). In frame III we observed pure soliton self-injection locking. In frame V the laser diode operated in a multi-frequency regime and this region can not be analyzed. If the laser doesn't switch to this regime, it would go out of the locked state and start to be free-running again. Then the current sweep will change its direction and the laser scans over the same resonance in the forward direction.

Action taken:

- We revised the caption of the Fig. 4 and gave the description of all Roman numbers.

4. The generated comb spectra is rather narrowband. There have been many demonstrations of broadband soliton microcomb generation in silicon nitride microresonators. Is this possible in an injection locked scheme? What, if any, are the limitations for achieving broad bandwidth?

Our reply:

First, the width of the comb spectrum depends on the microresonator group velocity dispersion (GVD, D_2). In our case, the GVD D_2 is large, which ultimately limits our soliton bandwidth. We can optimize this parameter in the future to obtain lower D_2 and wider comb. Nevertheless, the width of the presented soliton microcombs in the manuscript is comparable with conventional microcombs with low rep.rates [55].

Secondly, the comb width depends on the effective detuning $\zeta = 2(\omega_0 - \omega_{eff})/\kappa$ for a specific D_2 . In conventional soliton initiation setup the external laser with isolator is used and it's easy to change the laser detuning. The self-injection locking makes the laser generation frequency (ω_{eff}) locked to the microresonator (ω_0), so ζ is changing very slightly.

The main goal of our works was to understand how we change the effective detuning by changing the injection current. We have shown that not all values of the effective detunings are achievable (Fig. 1e). Our study shows ways to increase the effective detuning range and what parameters should be optimized. To get the broadest comb one should optimize Q-factor, the backscattering, and coupling.

Recently, broadband (near octave-spanning) soliton microcomb with 1-THz-FSR via self-injection locking has been demonstrated in NIST [36], where 1-THz-FSR microresonator is used. Conclusions of our work were used to explain the behavior of the device presented in this work.

[36] T. C. Briles et al., "Semiconductor laser integration for octave-span Kerr-soliton frequency combs," CLEO Conference 2020, STh1O.6, 2020

[55] Liu, J., Lucas, E., Raja, A.S. et al. Photonic microwave generation in the X- and K-band using integrated soliton microcombs. Nat. Photonics (2020).

Action taken:

- We added the following text to the "V. Conclusion and discussion" (page 9): "Therefore, the soliton self-injection locking provides, first, laser diode stabilization, second, microcomb generation, third, ultra-low noise photonic microwave generation. The main problem of this technique is the limitation of achievable effective detunings: a single-soliton state with a large detuning and broad bandwidth may be hard to obtain in the SIL regime. Further careful parameter optimization is needed for the comb bandwidth enhancement. One possible solution may be based on the fact that backscattering plays a major role and different schemes with increased backscattering may extend the range of the effective detunings in the locked state."
- We added the following to the Methods "Comb generation in SIL regime" (page 9): "Therefore, a single-soliton state with a large detuning and broad bandwidth may be hard to obtain in the SIL regime and further careful parameter optimization is needed."

REVIEWERS' COMMENTS

Reviewer #1 (Remarks to the Author):

I have read the responses carefully and the authors have responded well to the main reservation of my review. The revised manuscript also addresses most of my initial concerns and the drawn conclusions are now supported by more data including time-domain FROG measurements of self-injection locked solitons. I think the paper is now acceptable for publication to Nature Communications.

Reviewer #3 (Remarks to the Author):

This is a resubmitted manuscript on the theoretical and experimental investigation of self-injection locking of a DFB laser to a nonlinear microresonator for soliton microcomb generation. I appreciate the efforts that the authors have put in to revise and add content to the manuscript. I however still have reservations about the novelty of this work.

As mentioned in the previous review, this is not the first work demonstrating injection locking with some of the authors already publishing prior work in Nature Communications (Raja, et al., "Electrically pumped photonic integrated soliton microcomb," Nat. Commun. 10, 680 (2019).), and this work represents more details on the process. I recognize that the theoretical work is nontrivial and the microcomb community indeed will value this work. However, it is of my opinion that the work in its current form lacks novelty to warrant publication in Nature Communications which has a scope that is broader than just photonics. The authors do suggest that this device can be promising for use in data-centers or scientific instrumentation but their demonstration shows quite low powers (1.4 mW) which seems to imply that the power in each comb line is extremely low for a 30 GHz microcomb. Perhaps the manuscript can be further considered if the authors can project theoretically and add analysis on how to extend the bandwidth or increase the output powers in an injection locked system to be able to accommodate such applications.

Dear Reviewers,

First of all, we would like to express our deep appreciation to the reviewers, for the time they took to carefully read our manuscript, for a positive assessment of our work and for the valuable comments that help us improve the quality of presentation of our results.

In the following, we will respond in great detail (in black) to the reviewers' questions (in blue), point-by-point, as well as the action taken (in red). All changes in the revised manuscript are colored in blue.

Reviewer #1 (Remarks to the Author):

I have read the responses carefully and the authors have responded well to the main reservation of my review. The revised manuscript also addresses most of my initial concerns and the drawn conclusions are now supported by more data including time-domain FROG measurements of self-injection locked solitons. I think the paper is now acceptable for publication to Nature Communications.

Our reply:

We appreciate that the reviewer provided such a good idea to conduct time-domain FROG measurements and prove that self-injection locked soliton provide the same level of time coherence of convenient Kerr solitons. This measurement has been done for the first time, we expect that it will gain the manuscript impact to the community and will raise interest of readers.

Reviewer #3 (Remarks to the Author):

This is a resubmitted manuscript on the theoretical and experimental investigation of self-injection locking of a DFB laser to a nonlinear microresonator for soliton microcomb generation. I appreciate the efforts that the authors have put in to revise and add content to the manuscript. I however still have reservations about the novelty of this work.

As mentioned in the previous review, this is not the first work demonstrating injection locking with some of the authors already publishing prior work in Nature Communications (Raja, et al., "Electrically pumped photonic integrated soliton microcomb," Nat. Commun. 10, 680 (2019).), and this work represents more details on the process. I recognize that the theoretical work is nontrivial and the microcomb community indeed will value this work. However, it is of my opinion that the work in its current form lacks novelty to warrant publication in Nature Communications which has a scope that is broader than just photonics. The authors do suggest that this device can be promising for use in data-centers or scientific instrumentation but their demonstration shows quite low powers (1.4 mW) which seems to imply that the power in each comb line is extremely low for a 30 GHz microcomb. Perhaps the manuscript can be further considered if the authors can project theoretically and add analysis on how to extend the bandwidth or increase the output powers in an injection locked system to be able to accommodate such applications.

Our reply:

First of all, we thank the reviewer for the positive comments. We also appreciate that the reviewer has valued our effort to improve the manuscript. We would like to emphasize the novelty of our work and show the differences between our work and the rest.

Experimental and theoretical study of Kerr soliton generation via self-injection locking in nonlinear and active photonic circuits was our main motivation for this research. The first proof of concept [1] demonstrated that soliton microcomb may be developed with a hybrid

integration of III-V laser diode and CMOS-compatible silicon nitride photonic chip. In this work we focused on the fundamental principles of soliton self-injection locking, which has been never studied before despite the commercial use of OEwaves company and has been proved to be a demanded industrial product (Prism Award 2013 in the Defense and Security Category) [2].

Our finding has been already used in one of the most impressive microresonator-based projects - Direct On-Chip Digital Optical Synthesizer (DODOS). The group of Scott Papp in NIST studied the behavior of near-octave spanning self-injection locked microcomb and described it using our theoretical model looking for ways to broaden it [3].

Speaking about the practical issues of our manuscript we studied the dynamics of the system and didn't optimize all parameters of the experimental setup to achieve the highest power efficiency. Matching of the laser diode optical mode and the photonic chip bus waveguide's one is an essential task which allows to decrease total losses of the system. In a series of proof-of-concept experiments, it was shown that it's possible to connect InP lasers to passive silicon photonic circuits with insertion losses down to 0.4 dB [4].

Another important step is the optimization of the microresonator and the bus waveguide geometry in order to achieve maximum soliton power. The analytical expression for the line power, in Watts, of the DKS frequency comb at the output of the microresonator, solved in the limit of low resonator loss, is given by [5]:

$$P_{\mu} = \eta \kappa \frac{n_{\text{eff}} n_{\text{eff}_c} D_2 V_{\text{eff}}}{4 \omega c n_2} \text{sech}^2 \left(\frac{\pi}{2} \sqrt{\frac{D_2}{\kappa \zeta}} \mu \right),$$

where n_{eff} and n_{eff_c} are effective refractive indices of the WGM and the coupler mode, n_2 is the nonlinear index of the microresonator, D_2 is the second order dispersion (or GVD) coefficient, η is the pump coupling coefficient, κ and ω are the loaded linewidth and eigenfrequency of the WGM and V_{eff} is its effective volume. The μ is the mode number, counting from the pumped mode and the formula is valid only for $\mu \neq 0$ (the zero mode should take the interference with the pump into account). The ζ is the effective detuning. The WGM cross section area is assumed the same as for the output beam.

This formula provides insight into both the comb width and comb tooth power. First, the power can be increased by increasing the effective detuning ζ , which also leads to the comb width increase. This parameter has finite borders (see Eq. 1 in the main text) in the unlocked state but the maximum can be increased with the input power. However, its change is difficult in the locked state as the effective detuning is fixed and exhibits a complicated dependence on the pump power, microresonator backscattering rate and quality factor. Second, the sideband power can be increased by means of dispersion engineering. It is worth noting, that the optimum over D_2 exists, so if the power is important, it should not be decreased abruptly. Last, the coupling increase also allows to increase the sideband power. However this can decrease the loaded quality factor, increasing the power threshold for the comb generation.

To maximize the power of the weakest comb lines, the group velocity dispersion (GVD) and the coupling rate of the microresonator to the bus waveguide may be varied within values that can be engineered by varying the Si₃N₄ waveguide dimensions as well as the resonator-to-waveguide distance in combination with specially designed coupling regions [6].

Another approach for power enhancement is the use of normal GVD regime, which supports formation of dark soliton pulses. These temporal localized structures provide much higher pump-to-comb conversion efficiency [7] up to 41% [8]. The proposed in the main text theoretical model of nonlinear self-injection locking is valid for this regime. Moreover, our recent numerical studies suggest that the SIL allows solitonic pulse generation in normal GVD regime without any additional efforts [9].

Nevertheless, we demonstrated the self-injection locked soliton with power of 1.4 mW. The comb optical spectrum (Fig. 3b of the main text) consists of more than 30 lines with power of -20 dBm. The one of the record transmission data experiments, where transmission of a data stream of more than 50 terabits per second, utilizes the Kerr microcomb with 179 lines with power higher than -20 dBm [6].

[1] Raja, A. et al., "Electrically pumped photonic integrated soliton microcomb," *Nat. Commun.* **10**, 680 (2019).

[2] [spie.org/news/spie-professional-magazine-archive/2013-january/prism-award-finalists](https://www.spie.org/news/spie-professional-magazine-archive/2013-january/prism-award-finalists)

[3] Briles, T. C. et al., "Semiconductor laser integration for octave-span Kerr-soliton frequency combs," *CLEO Conference 2020, STh1O.6*, 2020.

[4] Billah, M. R. et al., "Hybrid integration of silicon photonics circuits and InP lasers by photonic wire bonding," *Optica* **5**, 876-883 (2018).

[5] Herr, T. et al. Temporal solitons in optical microresonators. *Nat. Photon.* **8**, 145–152 (2014).

[6] Marin-Palomo, P. et al. "Microresonator-based solitons for massively parallel coherent optical communications," *Nature* **546**, 274-279 (2017).

[7] Xue, X. et al., "Microresonator Kerr frequency combs with high conversion efficiency," *Laser Photon. Rev.* **11**, 1600276 (2017).

[8] Kim, B. Y. et al., "Turn-key, high-efficiency Kerr comb source," *Opt. Lett.* **44**, 4475 (2019).

[9] Kondratiev, N. M. et al., "Numerical modelling of WGM microresonator Kerr frequency combs in self-injection locking regime", *Proc. SPIE* **11358**, Nonlinear Optics and its Applications 2020, 113580O (1 April 2020).

Action taken:

- We added the following text into the Discussion Section with further steps to increase the comb power: "Another important question is increase of the total comb power (see Supplementary Note 7. First, the optimization of the laser diode mode and the photonic chip bus waveguide mode matching is an essential task to decrease the total insertion loss of 7 dB and increase the pump power. Optimization of the parameters of the photonic chip waveguide, particularly speaking, the second-order dispersion D_2 and the comb coupling rate to the bus waveguide will lead to the better extraction of the comb lines out of the microresonator [10]. Moreover, bright dissipative Kerr solitons are well-studied structures exhibiting fundamental limitation of the pump-to-comb conversion efficiency. Utilization of the dark soliton pulses, which formation is possible in microresonators with normal GVD, provides Kerr microcombs with high power per each line [62,63]. Moreover, our recent numerical studies suggest that the SIL allows solitonic pulse generation in normal GVD regime without any additional efforts [64]."

- We added the following text into the Discussion Section to highlight the importance of our research for wide range of readers: “This result is significant for laser systems with strong optical feedback (such as low-noise III-V/Si hybrid lasers and mode-locked lasers), oscillator synchronization, and other laser systems beyond integrated microcombs. A related example microrings is made of quantum cascade active media [65]. Therefore, our findings are relevant not only for the integrated photonics community but for a wide range of specialists.”
- We added the Supplementary Note 7 to the Supplementary Materials, where we consider ways to increase the comb power.